# 3D Characterization of the Molecular Neighborhood of ^•^OH Radical in High Temperature Water by MD Simulation and Voronoi Polyhedra

**DOI:** 10.3390/ijms24043294

**Published:** 2023-02-07

**Authors:** Lukasz Kazmierczak, Joanna Szala-Rearick, Dorota Swiatla-Wojcik

**Affiliations:** Faculty of Chemistry, Institute of Applied Radiation Chemistry, Lodz University of Technology, 90-924 Lodz, Poland

**Keywords:** OH radical, MD simulation, Voronoi polyhedra, aqueous solution, high temperature water, molecular structure

## Abstract

Understanding the properties of the ^•^OH radical in aqueous environments is essential for biochemistry, atmospheric chemistry, and the development of green chemistry technologies. In particular, the technological applications involve knowledge of microsolvation of the ^•^OH radical in high temperature water. In this study, the classical molecular dynamics (MD) simulation and the technique based on the construction of Voronoi polyhedra were used to provide 3D characteristics of the molecular vicinity of the aqueous hydroxyl radical (^•^OH_aq_). The statistical distribution functions of metric and topological features of solvation shells represented by the constructed Voronoi polyhedra are reported for several thermodynamic states of water, including the pressurized high-temperature liquid and supercritical fluid. Calculations showed a decisive influence of the water density on the geometrical properties of the ^•^OH solvation shell in the sub- and supercritical region: with the decreasing density, the span and asymmetry of the solvation shell increase. We also showed that the 1D analysis based on the oxygen–oxygen radial distribution functions (RDFs) overestimates the solvation number of ^•^OH and insufficiently reflects the influence of transformations in the hydrogen-bonded network of water on the structure of the solvation shell.

## 1. Introduction

Interest in processes involving aqueous ^•^OH radicals (^•^OH_aq_) results from the important role they play in biological systems, atmospheric chemistry, industrial processes, green chemistry technologies, and waste water treatment ([1,2,3], pp. 352–353). Reactions of this unselective oxidant with organic and inorganic compounds became a target of scientific investigations of desirable oxidation processes in biochemistry, remediation of soils and water, as well as harmful cell damage in biological systems, corrosion, and material degradation. Both the development of environmentally friendly technologies and suspension of undesirable reactions require knowledge of the behavior of ^•^OH radicals in aqueous environments at ambient up to supercritical temperatures. The latter conditions are relevant for technological applications. The past two decades have yielded numerous computational [4,5,6,7,8,9,10,11,12,13,14,15,16,17,18,19] and experimental [20,21,22,23,24,25,26,27] investigations. Computations involving both classical and ab initio simulations mainly focused on the solvation structure of ^•^OH_aq_ at ambient conditions, while the majority of experimental works have provided kinetic data obtained by means of pulse radiolysis technique for the temperature range of importance for the nuclear power industry and green chemistry technologies using high temperature water. Although the behavior and reactivity of ^•^OH_aq_ may strongly depend on the molecular vicinity, computational reports on its structural and dynamical properties at high temperatures are rather limited [8,13,14,17,18]. Moreover, the structural features were analyzed based on radial distribution functions (RDFs) describing site–site solute–solvent correlation in space in one dimension, i.e., along the radial coordinate. A main drawback of this 1D analysis is the loss of the angular correlations, and in the case of poorly defined RDFs, high inaccuracy of the determined structural features.

The aim of the present paper is to provide a three-dimensional (3D) coherent description of the solvation shell of ^•^OH_aq_ offered by the construction of the Voronoi polyhedron (VP) [28]. By definition, VP associated with a given central point is the minimal convex polyhedron whose planar faces are perpendicular bisector planes of lines connecting the center with the neighboring objects. Previously, we formulated a problem of VP construction to describe the molecular neighborhood of ^•^OH_aq_ at the body temperature [15]. A region delimited by the constructed VP unambiguously defines the space owned by ^•^OH in solution. At the same time, metric and topological properties of VP describe size and shape of the solvation shell providing more accurate estimates for a free volume and solvation number, i.e., the number of solvent molecules in the closest vicinity to a solute. The present paper employed the same methodology to study the molecular neighborhood of ^•^OH at several thermodynamic states, including ambient water, pressurized high-temperature liquid, and supercritical fluid. The methodology and considered thermodynamic conditions are described in Section 4. In Section 2, the statistical distributions of metric and topological features of the solvation shell are presented and compared with the RDF-based approach. In Section 3, the 3D characteristics of ^•^OH_aq_ are discussed in the context of hydration mechanisms indicated in our earlier studies [13,14,17,18].

## 2. Results

The constructed VP, by definition, describes the nearest molecular neighborhood of the radical known as a solvation shell or, alternatively, as a hydration shell in the case of an aqueous solvent. The calculated metric and topological features of VP include a spherical cage radius and face-weighted radius, the number of faces, volume, and surface area, and an asphericity factor (Section 4.3). These properties describe the size and symmetry of the solvation shell, the space owned by ^●^OH in solution (a free volume), and the solvation number. The Voronoi polyhedra were constructed using position coordinates of the oxygen atoms of solvent components taken from the simulated thermodynamic states listed in Table 1 (Section 4.1).

In Section 2.1, the influence of thermodynamic conditions is presented for average values (statistical means) of the metric and topological features of the constructed VP. The statistical distributions of these properties are analyzed in Section 2.2. In Section 2.3, the spatial extent of solvation shells and the solvation numbers resulting from the VP-based approach are compared with the assessments obtained from the analysis of the O_r_–O_w_ RDFs.

### 2.1. 3D Characteristics of ^•^OH_aq_

Figure 1 shows how the cage radius *R* and the number of faces *N*_F_ of the constructed polyhedra evolve with the thermodynamic conditions specified by temperature and density (Table 1). The radius *R* defines the spherical neighborhood of the radical oxygen atom (O_r_) and encompasses all the O_w_ atoms that make up the minimal-volume VP. A general trend observed for the statistical mean of *R* is to increase as water density decreases. The increase in the size of the average solvation shell is associated with the decrease in *N*_F_, corresponding to the solvation number. A particularly significant effect of density on *N*_F_ can be observed between 373 and 473 K, and on *R* at the supercritical temperatures. The diminishment of the solvation number is connected with significantly increasing volume *V* and surface area *S*_tot_ of the solvation shell (Figure 2). However, the rapid increase in the size, surface area and volume of the solvation shell seen for the supercritical region results in a relatively small change in *N*_F_. Such behavior indicates the high sensitivity of a shape of the solvation shell to fluid density.

Topological features were described here by face-weighted radius *R*_w_ and asphericity factor α, defined in Section 4.3. These parameters take into account the spatial distribution of H_2_O molecules within the solvation shell: *R*_w_ by the weighting factor *S*_i_/*S*_tot_ that enhances the contribution of molecules closer to ^•^OH, and asphericity factor α describing deviation from the spherical shape. The response of the topological parameters to the thermodynamic conditions is illustrated in Figure 3.

Up to 473 K the decrease in *R*_w_ is opposite to the behavior of the cage radius *R*, but consistent with the lowering of *N*_F_ (Figure 1). It suggests a stronger spatial correlation between ^•^OH and the closer H_2_O molecules. At the same time, higher values of the parameter α indicate greater asymmetry of solvation shells. At temperatures above 473 K, both radii show the same trend with decreasing density, although the discrepancy between *R* and *R*_w_ expands. A particularly high value of α obtained for low-density fluid means a highly asymmetric distribution of water molecules around ^●^OH.

### 2.2. Statistical Distributions of Metric and Topological Properties of ^•^OH_aq_

Histograms in Figure 4 describe the probability density distribution functions of *N*_F_, *R*_w_, and *V* of the constructed VP. Each panel depicts the histogram obtained for the solution at 310 K, the pressurized liquid at (573 K, 0.72 kg/L), and the supercritical fluid at (673 K, 0.550 kg/L). The statistical distributions of *N*_F_ calculated for the high temperature water are very similar but noticeably different from that obtained for 310 K. The influence of thermodynamic conditions on the histograms obtained for *R*_w_ and *V* is more pronounced. As the temperature increases and the density decreases, the statistical distributions become wider, more positively skewed, and shifted to the right.

In the supercritical region, the topology of the molecular neighborhood of the radical is largely determined by the fluid density. The dependence of the statistical distributions on density at 673 K is presented in Figure 5. At the lower density, the histograms of *R*_w_, and *V* are wider and right-shifted, whereas the histogram of *N*_F_ shifts to the left. With the decreasing density, all the distributions are more positively skewed.

Descriptive statistics of the probability distributions obtained for all the thermodynamic states have been demonstrated graphically using box-and-whisker and violin plots (Section 4.4). Figure 6 shows box-and-whisker plots for cage radius *R* and face-weighted radius *R*_w_. As can be seen, both radii are strongly correlated, but regardless of the thermodynamic conditions, *R*_w_ is smaller than *R*. A particularly large difference between means and medians was seen at 473 K. The interquartile ranges and length of the whiskers indicate that the distribution of *R*_w_ is less skewed and more concentrated. Thus, *R*_w_ more reliably describes the size of solvation shells of ^●^OH_aq_ than cage radius *R*. It is worth noting that since *R*_w_ is related to *V* and *S*_tot_ by Equation (3) and to the asphericity factor α by Equation (4), it combines metric and topological features.

In Figure 7 the box-and-whiskers plots for *R*_w_ are compared with the ones obtained for the asphericity factor α. The plots characterizing statistical distributions of *V* and *S*_tot_ are presented in Appendix A (Figure A1 and Figure A2). As can be seen, the distributions of *V*, *S*_tot_, and α are positively-skewed and platykurtic in the whole range of the investigated states. Despite the positive skew of these distributions, the *R*_w_ distributions obtained for liquid water below 400 K are slightly negatively skewed. At these conditions, the parameter α is approximately constant and close to unity (the mean 1.45–1.54 and median 1.42–1.49), indicating a roughly symmetric shape of the solvation shell.

A meaningful change in the statistics of all the VP features is observed at 473 K. The data obtained for (473 K, 0.881 kg/L) were poorly clustered, resulting in very diffuse probability distributions and noticeably deviating descriptive statistics. As discussed in Section 3, the observed misfits coincide with changes in the solvent structure occurring between 473 and 573 K.

The distributions of all the VP features obtained for high-temperature pressurized liquid and the supercritical fluid show positive skewness. The positive skewness is associated with the significant increase in α, meaning that the molecular neighborhood of ^•^OH is more diversified and the solvation shell loses a roughly spherical shape. At the same time, as Figure A1 and Figure A2 show, the volume available for ^•^OH within the solution noticeably increases.

### 2.3. RDF-Based Description Versus 3D Characteristics of Solvation Shell

The description of the solvation shell based on the construction of VP is more reliable because it takes into account the angular dependencies, which are averaged out in the RDF-based approach (Section 4.2). In Figure 8, the descriptive statistics of *R*_w_ are compared with the positions of the first maximum (*R*_max_) and the first minimum (*R*_m_) of the O_r_–O_w_ RDFs. As can be seen within a statistical uncertainty, both *R*_max_ and *R*_m_ are not very sensitive to the thermodynamic conditions. The position of the first minimum, which is the most commonly accepted estimate of the size of solvation shells, is clearly above *R*_w_. On the other hand, the spatial extent of the solvation shell described by *R*_w_ agrees well with *R*_max_. Particularly good accordance with the mean and the median was obtained for liquid water below 400 K, where the distribution of *R*_w_ fits the normal distribution (Figure A3). At (473 K; 0.881 kg/L) the interquartile range of *R*_w_ is broad and *R*_max_ falls into Q_3_. For the pressurized liquid above 500 K and the dense supercritical fluid, *R*_max_ falls into *Q*_1_. Some discrepancy between *R*_w_ and *R*_max_ seen for the low-density supercritical states results from the increasing asymmetry of the solvation shell, not reflected by *R*_max_.

Estimates of the solvation number resulting from the 1D and 3D approaches are confronted in Figure 9. In the RDF-based method, the solvation number is calculated by integration of the first peak to *R*_min_ (Section 4.2). Additionally, the number of faces is also compared with the results of RDF-integration to *R*_max_.

The mean values of *N*_F_ obtained for liquid water are in-between estimates resulting from integration of the O_r_–O_w_ RDF to *R*_max_ (down triangles) and to *R*_min_ (up triangles). However, with the increasing temperature and the decreasing density, the solvation number obtained by integration to *R*_min_ becomes closer to *N*_F_. It should be noted, however, that the mean and median of the *N*_F_ distribution do not significantly vary and show a different trend with the decreasing density of the fluid. The violin plots indicate that for liquid water at temperatures below 400 K the distributions of *N*_F_ well match the normal distribution. On the other hand, none of the distributions obtained for the supercritical fluid is as symmetric as the normal distribution.

## 3. Discussion

The changes in the metric and topological properties of the constructed VP polyhedra clearly show that the molecular neighborhood of ^•^OH is influenced by the transformations occurring in the structure of the highly associated solvent. According to the present view, in ambient conditions, water molecules form a gel-like structure [1]. Analysis of the hydrogen-bond connectivity patterns of pure solvent revealed that the continuous network comprises patches of four-bonded molecules and less ordered nets [29]. Patches become smaller with the increasing temperature and disappear at about 473 K. Below 400 K, when the solvent structure is stiffened by the presence of patches, the VP features (*R*, *R*_w_, *V*, *S*_tot_, *N*_F_, and α) do not change much. The solvation number (*N*_F_) and face-weighted radius (*R*_w_) undergo normal distributions, and the distributions of α indicate roughly symmetric solvation shells. These features are consistent with the localization of ^●^OH in cavities existing in the hydrogen-bonded water structure [12,14].

Simulation of pure solvent indicated the disappearance of patches and breakage of the continuous hydrogen-bond network into a variety of H-bonded clusters (smaller nets) between 473 and 573 K [29]. Influence of these transformations on ^●^OH_aq_ is reflected by very diffuse distributions of all the VP features and the discontinuous changes in the statistical parameters seen at 473 K.

The structure of high-temperature water (pressurized liquid and supercritical fluid) is less firmly defined. In consequence, the ^●^OH solvation shell loses its spherical shape, and becomes wider and more diversified. Our study on the dynamic properties of ^●^OH_aq_ showed that in high-temperature solvent ^●^OH is complexed with water via hydrogen-bonding interactions and moves as a molecular aggregate [18]. The formation of radical-water complexes was earlier recognized as the self-trapping mechanism of hydration and distinguished from the localization in cavities [13,14]. Here we have found that as the density of the fluid decreases, the span and asymmetry of the solvation sphere increase, whereas the solvation number remains almost constant. Stretching and sharpening of the constructed VP into one direction indicate that ^●^OH is directionally surrounded by the solvent, i.e., not covered with water molecules on some sides. Although the application of the VP method to the description of solvation shells in low-density fluid encountered limitations, the VP features confirm the directional accumulation of water molecules around ^●^OH in the supercritical fluid and are consistent with our conclusion that the radical-water complex acts as a center of condensation for the solvent molecules leading to minimization of Gibbs free energy of the system [13,14,18].

The structural changes in the molecular vicinity of ^•^OH can also be interpreted based on the concept of hydrogen bond (O:H-O) cooperativity and polarizability [30,31]. According to this concept, repulsion between lone pairs of the adjacent oxygen atoms (O: ↔ :O) couples the intermolecular O:H and the intramolecular H-O interactions to form hydrogen-bonded pair and drives the O:H-O bond to cooperativity. The tendency of the ^●^OH radical to replace H_2_O molecule in a broken hydrogen-bonded network and the directional solvation of ^•^OH are driven by repulsive O: ↔ :O interactions and reflect an inclination of the system to maintain cooperativity.

## 4. Methods

To investigate the structural features of the molecular neighborhood of ^●^OH_aq,_ we used classical MD simulation of an aqueous solution containing the hydroxyl radicals at high dilution. In total, we performed thirteen simulations: three for ambient and elevated temperatures, four for the pressurized liquid, and six for the temperature above 647 K, being the critical temperature of the water. Parameters of the simulated thermodynamic states of water are presented in Table 1. The adopted potentials and the simulation method, shortly summarized in Section 4.1, were described in detail elsewhere [12,13,14]. Stored configurations of the equilibrated systems were processed to obtain 1D and 3D characteristics of the solvation shells of the radical, as described in Section 4.2 and Section 4.3, respectively.

### 4.1. MD Simulation

The NVE ensemble simulations were carried out for the diluted aqueous ^•^OH solution. The system was modeled by the periodically repeated cubic box containing one ^•^OH radical and 400 H_2_O molecules. The size of the simulation box was calculated according to the density of the thermodynamic states listed in Table 1. All the solvent components were described by the flexible models: three-site central force potential [32] for water with partial charges +0.33 *e* and −0.66 *e* centered on the hydrogen (H_w_) and oxygen (O_w_) atoms of H_2_O molecule, leading to the critical point parameters (609.5 K, 0.378 kg/L, 27.5 MPa), and the compatible two-site model [12] for ^•^OH with the partial charges +0.375 *e* and −0.375 *e* centered on the radical hydrogen (H_r_) and the radical oxygen (O_r_), respectively. The employed model potentials include short-range pair interactions of hydrogen atoms to better describe spatial hindrance resulting from the presence of hydrogen-bonding interactions. The equations of motion were integrated using the Verlet algorithm and assuming the simulation step of 0.1 fs. Long-range and short-range non-bonding interactions were treated by the Ewald summation method and the shifted-force one, respectively. The pre-equilibration stage required ca. 4·10^6^ time steps. Lengths of the production runs varied from 50 to 110 ps. Positions and velocities of the molecular sites were stored every 1 fs. The stability of the total energy was 10^−6^ < Δ*E*/E < 10^−5^. The temperature fluctuation was within 10 K.

### 4.2. RDF-Based Approach

For the purpose of the present study the positions of molecular sites in the stored configurations were used to calculate O_r_–O_w_ partial RDF, $g_{O_{r}O_{w}}$. The $g_{O_{r}O_{w}}\left(r\right)$ function of the radial coordinate *r* presents the distribution of water–oxygen atoms around the radical–oxygen site. The first peak of $g_{O_{r}O_{w}}\left(r\right)$ defines the size of the solvation shell by the position of the maximum, *R*_max_ and the minimum *R*_min_. The integration of the first peak was used to estimate the number of solvent molecules constituting the solvation shell. The solvation number or alternatively hydration number in the case of the aqueous solution was calculated as a running integration number:(1)$$n_{O_{r}O_{w}}=4\pi \rho \int_{R_{0}}^{R_{min}}g_{O_{r}O_{w}}\left(r\right)dr$$
where *ρ* is the number density of the water oxygens and *R*_0_ marks the beginning of the first peak. The running integration number was also calculated assuming *R*_max_ for the upper integration limit in Equation (1).

### 4.3. VP-Based Approach

VP was constructed with respect to the oxygen atoms nearly allocating mass centers of the ^•^OH and H_2_O molecules. Position coordinates of the oxygen atoms O_r_ and O_w_ were taken from the stored configurations of the equilibrated system. The number of configurations analyzed for each thermodynamic state is given in Table 1. The construction method was detailed described in our earlier paper [15]. Each constructed VP is characterized by the numbers of faces (*N*_F_), edges (*N*_E_), and vertices (*N*_V_), related by the Euler relation:*N*_F_ + *N*_V_ − *N*_E_ = 2(2)

The coordinates of vertices were used to calculate: the surface area of faces *S*_i_ (i = 1, 2, …, *N*_F_), total surface area *S*_tot_, and volume *V* of the constructed VP. Mathematical details of these calculations are given in ref. [15]. The minimal-volume VP with the smallest number of faces was taken as the solvation shell of ^•^OH. Consequently, the number of faces of the minimal-volume VP was considered the solvation number. We defined cage radius *R* as the radius of the spherical neighborhood of the central O_r_ atom encompassing all the O_w_ atoms that make up the minimal-volume VP.

The topological properties of the minimal-volume VP were described by the number of faces *N*_F_, face-weighted radius *R*_w_, and asphericity factor α. Definition of *R*_w_ accounts for different O_r_–O_w_ distances of water molecules in the solvation shell. It follows from the VP construction that the O_w_ atoms closer to the center (O_r_) result in larger faces. The contribution of these molecules was enhanced by introducing the weighting factor *S*_i_/*S_tot_*. As shown previously [15], it leads to Equation (3) relating *R*_w_ with *V* and *S*_tot_ of the constructed VP:(3)$$R_{w}=\frac{3V}{S_{tot}}$$

Moreover, *R*_w_ is connected with the asphericity factor α defined by Equation (4):(4)$$\alpha =\frac{S_{tot}^{3}}{36\pi V^{2}}=\frac{V}{\frac{4}{3}\pi R_{w}^{3}}$$

Since for an ideal sphere α = 1, a deviation from unity is a measure of the asymmetry of the spatial distribution of H_2_O molecules within the solvation shell.

### 4.4. Statistical Analysis

Normalized probability density functions, or shortly after, probability distributions, were calculated for the metric parameters (*R*, *V*, *S*_tot_) and the topological features (*R*_w_, *N*_F_, α) of the constructed VPs. The probability distributions of the number of faces, face-weighted radius and VP-volume were presented in the form of histograms. In addition, the locality, spread, and skewness of the statistical distributions of all parameters were demonstrated graphically using box-and-whisker and violin plots [33,34].

The box-and-whisker plot displays the statistical mean (a point marker), the minimum (bottom whisker), the maximum (upper whisker), the median, the first (*Q*_1_), and third (*Q*_3_) quartiles. The median divides the box into the lower and upper subsections, and *Q*_1_ and *Q*_3_ are marked by the lower and upper subsections, respectively. In that way, the spacing in each subsection of the box-plot indicates the degree of dispersion and skewness of the data. Relative positions of the quartiles *Q*_1_, *Q*_3_, mean, and median provide information about the skewness and kurtosis of a given statistical distribution.

A violin plot is similar to box plot, except that it additionally shows the values of the probability density distribution function at the mean, *Q*_1_, median (*Q*_2_), *Q*_3_, maximum (*Q*_4_)_,_ and minimum (*Q*_0_).

## 5. Conclusions

Characteristics of the molecular neighborhood of ^•^OH in aqueous systems at conditions ranging from ambient to supercritical are needed for the mechanistic understanding of desirable and undesirable reactions initiated by this short-lived and unselective oxidant. In the present work, we used the classical MD simulation and the construction of Voronoi polyhedron to provide 3D description of ^•^OH solvation shell at several thermodynamic states of water, including the pressurized high temperature liquid and supercritical fluid. VP was constructed with respect to the oxygen atoms nearly allocating mass centers of the ^•^OH and H_2_O molecules. Position coordinates of the oxygen atoms were obtained from MD simulation of the diluted ^•^OH solution.

Statistical distributions of the solvation number, size, volume, surface area, and asphericity of the solvation shell were presented as a function of thermodynamic conditions using box-and-whisker and violin plots. A reliable estimate of the spatial extent of the solvation shell is provided by the face-weighted radius, which links the metric and topological features of the constructed VP. The dependence of these properties on the thermodynamic state of the solution reflects the interplay between solute–solvent and solvent–solvent interactions, and are consistent with the structural transformations occurring in the hydrogen-bonded network of water [29]. The cavity-localization in the solvent below 400 K is evidenced by a negligible change in metric and topological parameters of the constructed VP. Abrupt changes of these parameters at 473 K coincide with the breakage of the continuous network into clusters of hydrogen-bonded molecules, and a decisive effect of density on the VP properties calculated for sub-and supercritical conditions confirms the directional accumulation of water molecules around ^●^OH.

Finally, compared to the oxygen–oxygen RDF-based analysis, the VP-based approach provides more accurate data on the solvation numbers and the geometry of the solvation shells.

## Figures and Tables

**Figure 1 ijms-24-03294-f001:**
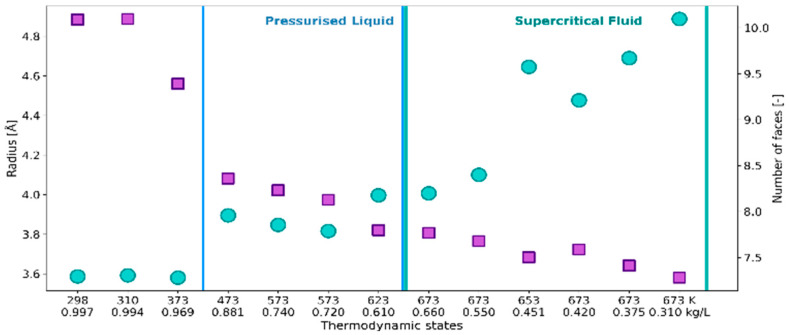
Cage radius *R* (circles, left axis) and the number of faces *N*_F_ (squares, right axis) of VP representing solvation shell of ^•^OH_aq_ in the thermodynamic states defined by temperature and density as in Table 1.

**Figure 2 ijms-24-03294-f002:**
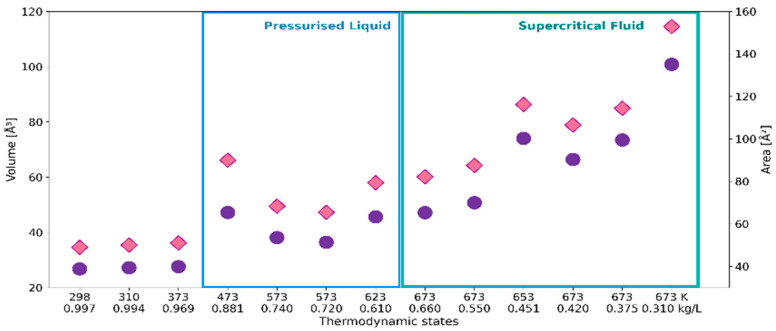
Volume *V* (circles, left axis) and surface area *S*_tot_ (diamonds, right axis) of VP representing ^•^OH_aq_ at the thermodynamic states defined by temperature and density as in Table 1.

**Figure 3 ijms-24-03294-f003:**
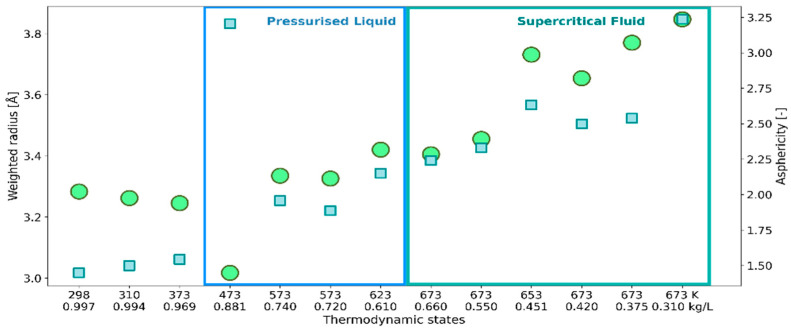
Face-weighted radius *R*_w_ (circles, left axis) and asphericity factor α (squares, right axis) of VP representing ^•^OH_aq_ at the thermodynamic states defined by temperature and density.

**Figure 4 ijms-24-03294-f004:**
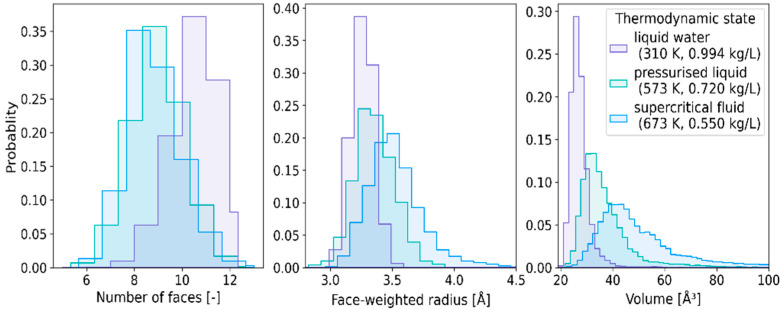
Histograms representing statistical distributions of the number of faces (**left**), face-weighted radius (**central**), and VP-volume (**right**) at the thermodynamic conditions specified in the right panel.

**Figure 5 ijms-24-03294-f005:**
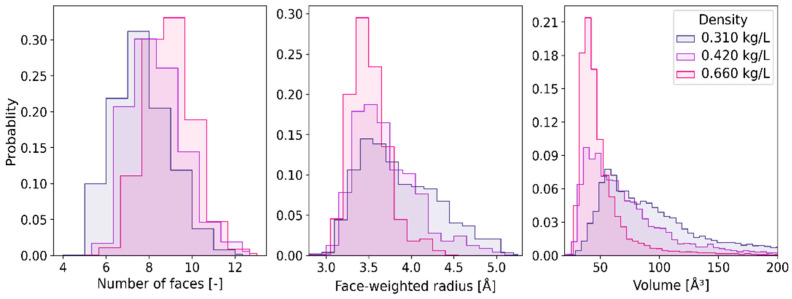
Histograms representing the effect of density on the statistical distribution of the number of faces (**left**), face-weighted radius (**central**), and VP-volume (**right**) at the supercritical temperature 673 K.

**Figure 6 ijms-24-03294-f006:**
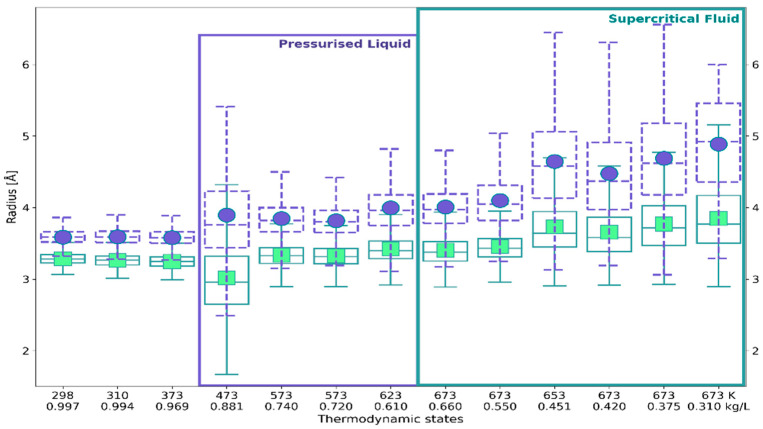
Box-and-whiskers plots displaying median, 1st and 3rd quartiles, minimum (bottom whisker), maximum (upper whisker), and means (points) of statistical distributions of cage radius R (circles, dashed contours) and face-weighted radius *R*_w_ (squares, solid contours).

**Figure 7 ijms-24-03294-f007:**
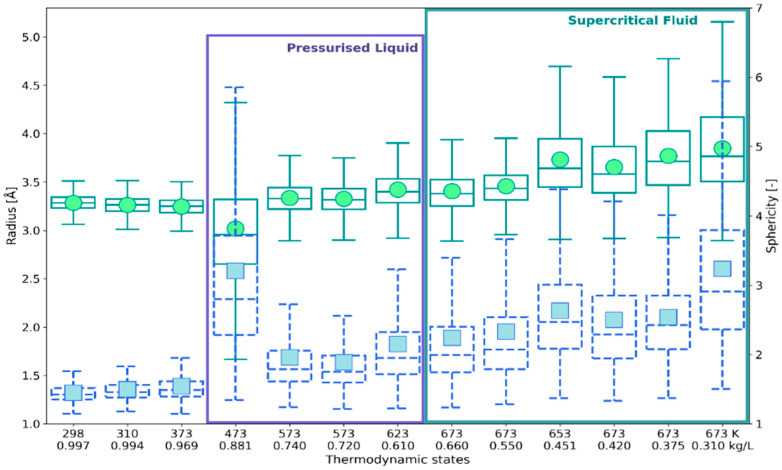
Box-and-whiskers plots of face-weighted radius *R*_w_ (circles, solid contours, left axis) and asphericity factor α (squares, dashed contours, right axis).

**Figure 8 ijms-24-03294-f008:**
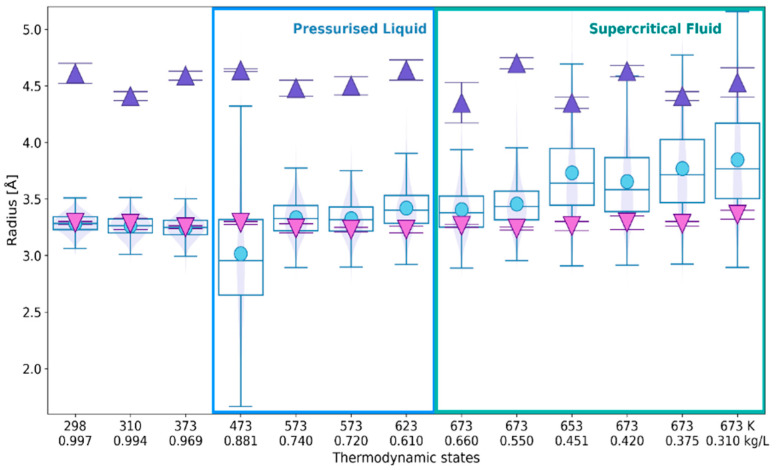
Face-weighted radius *R*_w_ (circles in box-and-whiskers-violin plot) versus RDF-based assessment of spatial extent of the ^•^OH_aq_ solvation shell: (down triangles) 1st maximum position of O_r_–O_w_ RDF, (up triangles) position of the 1st minimum. The thermodynamic states are defined in Table 1. The values of the probability density distribution function are shown by the violin plots.

**Figure 9 ijms-24-03294-f009:**
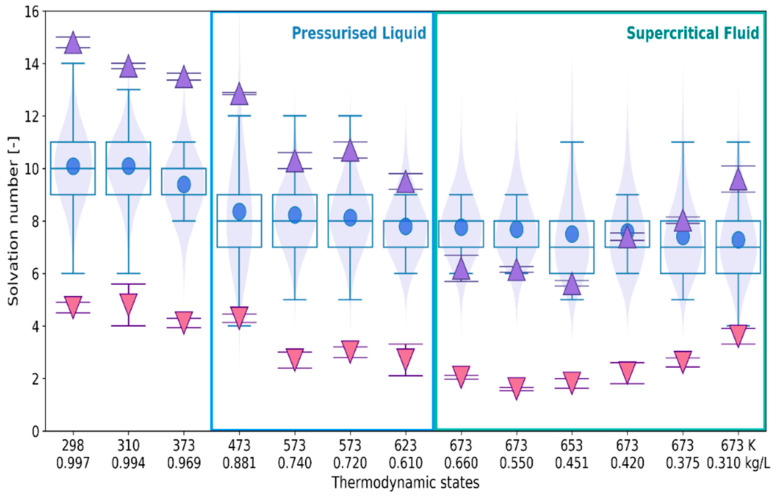
The number of faces *N*_F_ (circles in box-and-whiskers-violin plot) versus RDF-based assessment of solvation number of the ^•^OH_aq_: by integration of the O_r_–O_w_ RDF to the 1st maximum (down triangles), to the 1st minimum (up triangles). The thermodynamic states are defined by temperature and density as in Table 1. The violin plots additionally show the values of the probability density distribution function.

**Table 1 ijms-24-03294-t001:** The parameters of the simulated solutions of the hydroxyl radical in water: temperature, density, and the number of configurations analyzed for each thermodynamic state.

Temperature [K]	Density [kg/L]	Number of Probes
298	0.997	24,996
310	0.994	25,000
373	0.969	24,999
473	0.881	21,036
573	0.740	24,493
573	0.720	24,843
623	0.610	24,577
673	0.660	24,376
673	0.550	24,353
653	0.451	22,613
673	0.420	23,020
673	0.375	19,929
673	0.310	19,358

## Data Availability

Data is contained within the article and Appendix A.

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
