# Peer review of "3D Characterization of the Molecular Neighborhood of •OH Radical in High Temperature Water by MD Simulation and Voronoi Polyhedra"

_ijms, 2023, doi:10.3390/ijms24043294_

Round 1
Reviewer 1 Report
The paper deals with solvation of HO- hydrogen oxide using MD simulations, which give a different views that could be interesting. Recent progress show that HO- prefers the site of replacing an H2O in the hydrogen bonding net work. The introduction of excessive lone pairs derive the O::O super-hydrogen bond that exerts a force to neighboring O:H-O bond. This work complements the understanding, so I recommend acceptance with commenting on the O:H-O transition to O::O repulsion.
Authors are recomended the following:
[1] The physics behind water irregularity, Physics Reports, 998 (2023) 1-68.
[2] Unprecedented O:⇔:O compression and H↔H fragilization in Lewis solutions, PCCP, 21 (2019) 2234-2250.
Reviewer 2 Report
The manuscript is focused on the study of OH radical in aqueous conditions ranging from ambient to supercritical. The classical MD simulations and the Voronoi polyhedron to give a 3D description of OH radical solvation shell at different temperature are used. Results demonstrated as the topological properties of VP polyhedral strongly depends on the molecular neighborhood of OH radical and strictly correlated to the ambient conditions. The hydrogen-bond network is considered at different temperature and its role on the VP features is highlighted. In addition, a comparison with the more standard RDF-based analysis is reported. The manuscript is well-written and the conclusions are fully support by numerical experiments. Thus, I suggest the publication in the present form.
Author Response
We would like to thank the referee for the nice feedback. As suggested we have tried to improve the style and corrected typos. All revisions made to the manuscript have been highlighted.
Reviewer 3 Report
•OH is very important in biological systems, atmospheric chemistry, industrial processes, green chemistry technologies, and waste water treatment. This manuscript reports the classical molecular dynamics (MD) simulation and the technique based on construction of Voronoi polyhedra were used to provide 3D characteristics of the molecular vicinity of the aqueous hydroxyl radical. The decisive influence of the water density on the geometrical properties of the •OH solvation shell in the sub- and supercritical region: with the decreasing density, the span and asymmetry of the solvation shell increase. The 1D analysis based on the oxygen-oxygen radial distribution functions (RDFs) over-estimates the solvation number of •OH and insufficiently reflects the influence of transformations in the hydrogen-bonded network of water on the structure of solvation shell. The results are very good and interesting.
Author Response

(The authors gave the same response as above.)
